# Efficient single-photon emission via quantum-confined charge funneling to quantum dots
Sanghyeok Park[1,2], Khalifa M. Azizur-Rahman [1,2], Darryl Shima[3], Ganesh Balakrishnan[3], Jaeyeon Yu[1,2], Hyunseung Jung[1,2], Jasmine J. Mah [1,2], Samuel Prescott [4], Pingping Chen[2,5], Sadhvikas Addamane[1,2], Douglas Pete[1,2], Andrew Mounce [1,2], Ting Shan Luk[1,2], Prasad P. Iyer [1,2], Igal Brener [1,2] & Oleg Mitrofanov [4] ✉

Quantum light sources, particularly single-photon emitters (SPEs), are critical for quantum communications and computing. Among them, III-V semiconductor quantum dots (QDs) have demonstrated superior SPE metrics, including near-unity brightness, high photon purity, and indistinguishability, making them especially suitable for quantum applications. However, their overall quantum efficiency—determined by a product of the internal, excitation, and outcoupling efficiencies—remains limited, primarily due to low (typically below 0.1%) excitation efficiency. To mitigate the low efficiency under non-resonant pumping, here we realize liquid droplet etched GaAs QDs in a microscale 3D AlGaAs charge-carrier funnel. The funnel channels charge carriers to the QD and enhances the overall emission efficiency by over one order of magnitude while preserving the SPE behavior. We reveal that a modified energy landscape around the QD leads to the excitation efficiency improvement. These energy landscape-modified QDs can be operated with optical excitation up to 10 μm away, raising the promise of efficient electrically driven QD SPEs for quantum information systems.

Quantum light sources and specifically single-photon emitters (SPEs) have become the key enabling component for quantum communications and quantum computing[1–4]. A multitude of SPEs ranging from cold atoms[5,6] to defects in solids[7–9] and 2D materials[10,11] have been developed to provide the single-photon source functionality. Among them, III-V group semiconductor quantum dots (QDs)[12,13] represent one of the most established SPEs, and their use has evolved recently from initial proof-of-principle demonstrations (e.g., generation of photon-number states[14] and cluster states[15,16]) to practical deployment in quantum computing processors[17].

Semiconductor QD-based SPEs have shown some of the best metrics for quantum computing[1,4]: close to unity brightness[18–20], high photon purity[14,16,17,20,21] and near-unity indistinguishability[16,17,19,20,22]. They also hold high potential for scalability and integration, and for on-demand electrical driving[23–27]. However, the best SPE performance so far has relied on resonant optical pumping of QDs, which requires sophisticated filtering techniques[28–34] and highly fine-tuning of the excitation laser, limiting the scalability of the approach. Non-resonant optical pumping and, ultimately,

electrical driving simplify the filtering of photons, but these approaches tend to generate parasitic charge carriers and phonons near the QD, leading to faster dephasing and deterioration of the photon indistinguishability[4,12]. Furthermore, the overall quantum efficiency, i.e., the number of pump photons or injected charge carriers required to produce a photon in the single-photon state, is far from the desired value of unity.

Here, we demonstrate a solution to the low overall QD efficiency problem by developing a material-based microscale 3D charge-carrier funnel for QDs grown by molecular beam epitaxy (MBE). In a single MBE process, we realized a unique combination of an in situ nanoscale GaAs QD and a microscale AlGaAs charge-carrier funnel. The funnel forms an attractive potential, which channels the charge carriers injected within the microscale volume toward the embedded GaAs QDs, increasing the overall quantum efficiency by over one order of magnitude compared to ordinary QDs (o-QDs) in the same sample. Using nanoscale structural analysis and photoluminescence (PL) imaging, we reveal that the enhancement in efficiency originates from a modified energy landscape around the QD, which is

[1]Sandia National Laboratories, Albuquerque, NM, USA. [2]Center for Integrated Nanotechnologies, Sandia National Laboratories, Albuquerque, NM, USA. [3]Center for High Technology Materials, University of New Mexico, Albuquerque, NM, USA. [4]Electronic and Electrical Engineering, University College London, London, UK. [5]Electrical, Computer and Energy Engineering, University of Colorado Boulder, Boulder, CO, USA. ✉e-mail: o.mitrofanov@ucl.ac.uk

determined by the material composition and the effect of quantum confinement within the funnel. We demonstrate that these energy landscape-modified (ELM) QDs operate as SPEs, making them attractive as light sources for quantum applications. The ELM-QDs also enable a regime of remote QD excitation, where charge carriers are injected microns away from the QD, raising the potential for realizing efficient electrically driven on-demand SPEs built on a scalable platform[27,35].

## Results

For a single QD, the overall quantum efficiency $\eta$ is a product of the internal quantum efficiency $\eta_{int}$, the probability of emitting a photon when the QD is in the excited state, as well as the excitation and the outcoupling efficiencies, $\eta_x$ and $\eta_{oc}$: $\eta = \eta_{int}\eta_x\eta_{oc}$. The internal quantum efficiency $\eta_{int}$ is already close to unity for semiconductor QDs[36,37], while the outcoupling efficiency $\eta_{oc}$ can be engineered to approach unity by introducing cavities, resonators and metasurfaces[38–43], which modify the photonic environment around the QD and influence the outcoupling, directivity, as well as the spontaneous emission rate via the Purcell effect. In stark contrast, the excitation quantum efficiency $\eta_x$ typically stays below 0.1%, resulting in considerable losses of pump power during optical excitation[18–20]. Furthermore, compensation of the low excitation efficiency with higher pump power deteriorates the SPE properties due to the parasitic charge carriers and phonons near the QD.

The problem of low excitation efficiency is mitigated here using the concept of ELM-QD illustrated in Fig. 1a: the GaAs QD is located in a low aluminum (Al) fraction (~20% Al) AlGaAs micrometer-scale disk, which serves as a charge-carrier funnel within a higher Al fraction (~40% Al) AlGaAs barrier layer. The bandgap energy in the disk is lower than the bandgap energy in the barrier, varying gradually between the two levels due to the quantum confinement effect (Fig. 1b). Charge carriers within a microscale volume surrounding the QD, therefore, can be stored and funneled to the QD.

To realize this concept using MBE, where the formation of 3D structures, such as QDs, quantum rings[44,45] and nanowires[46], requires special growth conditions, we introduce a process of Ga droplet crystallization which forms an AlGaAs disc, just before the growth of QDs. First, during the growth of the $Al_{0.4}Ga_{0.6}As$ barrier layer, a Ga droplet is deposited on its surface (Fig. 1c). The droplet then crystallizes into AlGaAs by reacting with incoming aluminum and arsenic (As). Due to the energetics of the MBE growth process, AlGaAs exhibits preferential crystallization along (1$\bar{1}$0) directions, transforming the droplet into a microscale AlGaAs disk[47]. The Al composition in the disk is ~20%, lower than that in the barrier layer. After the disk has fully crystallized, GaAs QDs are grown using the liquid droplet etched (LDE) process with Al droplets, with only a few dots being etched into and grown inside the AlGaAs disk. Finally, the disk and the QDs are covered with a barrier layer of $Al_{0.4}Ga_{0.6}As$ (Fig. 1c, see Supplementary Note 1).

As a result of this growth process, we form QDs in two energetically different environments: o-QDs are located in the $Al_{0.4}Ga_{0.6}As$ barrier and

ELM-QDs are located inside the sparsely distributed $Al_{0.2}Ga_{0.8}As$ disks, serving as the charge-carrier funnels illustrated in Fig. 1c. The difference between the two kinds of QDs is striking, as evident in PL images of the sample, where the ELM-QDs show over one order of magnitude higher photon emission compared to the o-QDs (Fig. 1d).

To reveal the nanoscale structure of the AlGaAs charge-carrier funnel containing the QDs, we examined them with cross-sectional transmission electron microscopy (TEM). In Fig. 2, four layers can be distinguished in a TEM image of a representative funnel (Fig. 2a), thanks to the clear contrast in brightness caused by variation in the Al composition between the layers. The bottom $Al_{0.7}Ga_{0.3}As$ sacrificial layer has the darkest shade due to the highest Al content. Above it is a lighter gray ~30-nm-thick barrier $Al_{0.4}Ga_{0.6}As$ layer. The AlGaAs funnel, which has the lightest tone corresponding to the lowest Al content, is located above it. The thickness of this layer varies in the image from 10 to 20 nm at the edges to approximately 150 nm at the center of the funnel. Finally, another medium-gray $Al_{0.4}Ga_{0.6}As$ barrier layer caps the heterostructure. The funnel is fully contained within the $Al_{0.4}Ga_{0.6}As$ barrier. An atomic force microscopy topographical image of an area near a typical funnel is shown in Fig. 2d. It displays a raised disk of ~8 μm in diameter, with a smaller partial ring-shaped ridge (~1–2 μm) at the center.

Higher magnification TEM images show the evidence of LDE GaAs QDs formed just above the ridges within the AlGaAs funnel (Fig. 2b). The QDs can be identified by thin dark streaks (high Al composition) only a few nm above the top interface of the AlGaAs ridges. The streaks appear where the TEM section sliced through thin Al 'puddles,' which form around QDs[48], whereas the QDs themselves are not visible (Supplementary Note 2). Nevertheless, since LDE QD are located below the puddles (Fig. 1b), the QDs are likely to be inside the AlGaAs charge-carrier funnel, ~1-10 nm below the streaks. We note that the LDE QDs are preferentially formed on the ridge slopes, close to the region where the $Al_{0.2}Ga_{0.8}As$ disk thickness is highest. This occurs due to the growth energetics of LDE QDs, which preferentially form along the (1$\bar{1}$0) facets[47].

The low Al composition in the funnel provides the lowest potential energy for charge carriers and excitons, and it therefore can attract them to the QD. To evaluate the corresponding energy landscape near the QDs quantitatively, we performed the energy dispersive X-ray spectroscopy (EDS) and mapped the elemental composition in and near the funnel, along line scans A and B (Fig. 2c). The Al composition in the AlGaAs barrier (blue regions) is ~40%, whereas it drops down to ~20% in the funnel (pink region). Using the funnel's elemental composition and its geometry, we calculate the conduction band energy and the quantum confinement energy within it (see Methods). The corresponding energy profile is shown in Fig. 2e; it resembles that of an in-plane potential well with a parabolic profile. Although the composition within the funnel is uniform, the confinement energy for electrons increases gradually near the edges, where the disk thickness decreases. The energy eventually approaches the conduction band

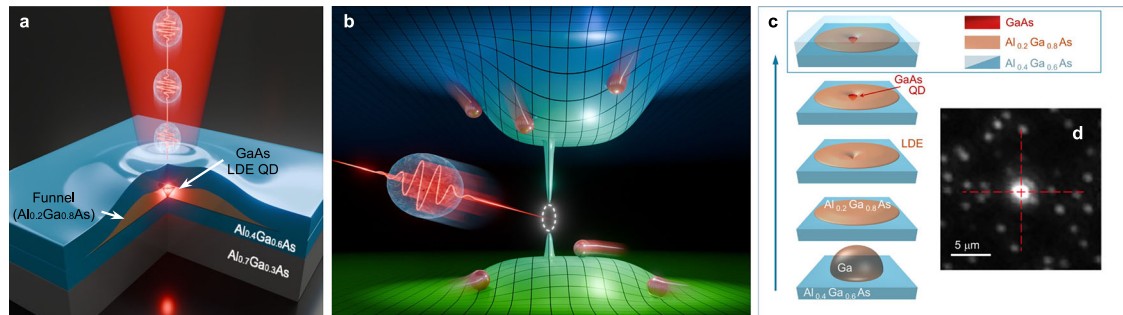

**Fig. 1 | Quantum dot in charge-carrier funnel. a** Schematic of ELM-QD: an $Al_{0.2}Ga_{0.8}As$ disk (orange) acting as a charge-carrier funnel, ~5–10 μm in diameter, with a GaAs QD near the center (red). The disk attracts photoexcited charge carriers toward the QD inside the disk, increasing the efficiency of the QD. **b** Illustration of the conduction (top) and valence (bottom) band energy landscapes near the QD with electrons and holes funneled towards it. **c** Schematic diagram of the ELM-QD formation process during MBE growth. **d** Photoluminescence image of a sample area containing an ELM-QD (bright) at the center and o-QDs.

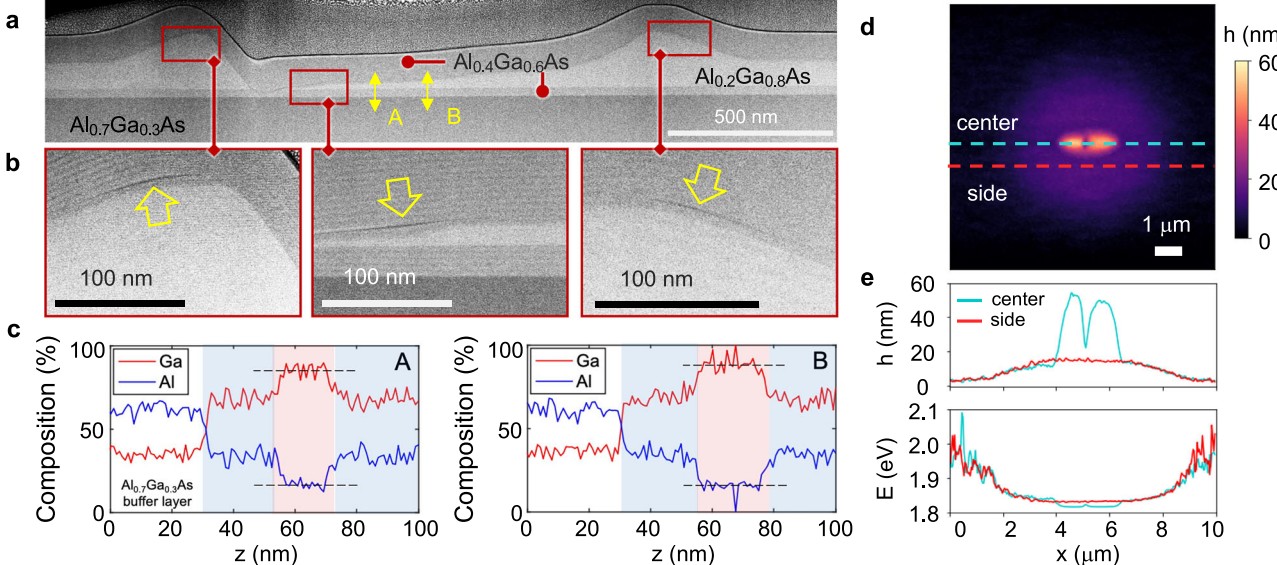

**Fig. 2 | Nanoscale structure of ELM-QDs. a** Cross-sectional TEM image of a representative single $Al_{0.2}Ga_{0.8}As$ funnel at ×80,000 magnification showing the central area of the disk and two sides of the ring-shaped ridge. **b** Cross-sectional TEM images of LDE GaAs QDs at higher magnification (×150,000, QD location in (**a**) is indicated with red boxes). Dark streaks (indicated with yellow arrows) show cross-sections of Al puddles formed near the QDs. **c** Ga and Al composition along line scans A and B in (**a**) obtained with EDS. The Ga (Al) composition is shown with the red (blue) line. **d** Atomic force microscopy (AFM) image of a typical ELM-QD. **e** Topography and the corresponding bandgap energy profile along two AFM line scans are indicated by the blue and red dashed lines in (**d**). Bandgap energy in the funnel is calculated using the AlGaAs disk thickness profiles and the Al composition.

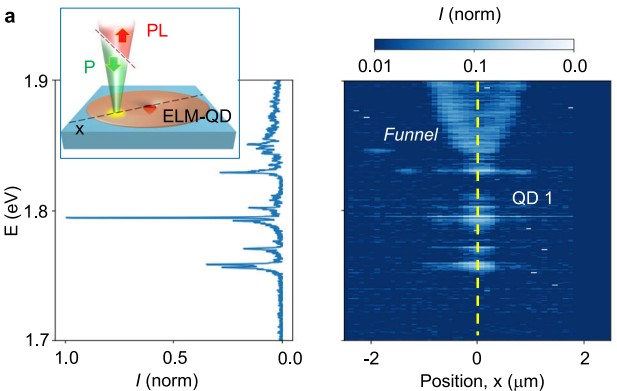

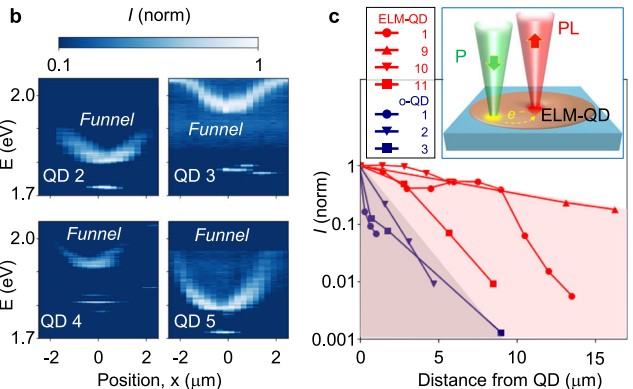

**Fig. 3 | Photoluminescence (PL) signatures of ELM-QDs. a** Right panel: spatial distribution of PL for a representative ELM-QD (ELM-QD 1). Micro-PL spectra were measured using an optical excitation beam of ~1 μm in diameter translated across the surface of the QD sample as illustrated in the Inset. The QD position is indicated in the map with a vertical yellow dashed line, and the corresponding micro-PL spectrum is shown in the Left panel. **b** Micro-PL maps of additional four ELM-QDs. **c** Remote optical excitation of ELM-QDs (illustrated in the Inset): normalized PL intensity as a function of the distance from the point of excitation for ELM-QDs (red lines) and o-QDs (dark blue lines).

level in the barrier. A similar energy landscape is experienced by the holes. As a result, the photoexcited charge carriers are attracted and funneled toward the funnel's central area, where QDs are preferentially formed, close to the ridges (the lowest potential energy). We note that the quantum confinement energy for an $Al_{0.2}Ga_{0.8}As$ layer thicker than ~50 nm is small, <1.7 meV, and it only weakly depends on the thickness. It therefore makes the energy landscape relatively flat in the central area of the funnel (Fig. 2e).

The energy landscape in the vicinity of the QDs becomes experimentally evident in a PL spectral map measured along a line scan through an ELM-QD (Fig. 3a): at the map center is the QD with a series of spectral lines corresponding to QD exciton states (left panel). In the energy region above the QD exciton emission, one can observe a broad PL peak continuously changing its energy along the line scan, with the lowest energy point close to the QD location. As the distance from the QD increases, the peak gradually shifts to higher energies, following an inverted bell-shaped trace in the map. This PL peak represents the bandgap energy within the funnel. As the

$Al_{0.2}Ga_{0.8}As$ disk becomes thinner towards the edges, the PL peak energy shifts to the exciton energy in $Al_{0.4}Ga_{0.6}As$ (Supplementary Note 3), and the shift is consistent with the calculations of the quantum confinement energy within the funnel (Fig. 2e).

## Discussion

This modified energy landscape facilitates the capture and channeling of photoexcited charge carriers in its vicinity of the QD. We found that all investigated ELM-QDs displayed PL maps similar to the map in Fig. 3a, with some variation in the minimum bandgap energy. Examples of the PL spectral map are summarized in Fig. 3b and in Supplementary Note 3: the minimum exciton energy at the funnel center varied from ~250 meV to ~50 meV below the exciton energy in the barrier. The elemental uniformity observed in the EDS measurements (Fig. 2) suggests that the thickness variation, rather than a change in material composition, is the major factor defining the energy landscape. The disk thickness and the corresponding

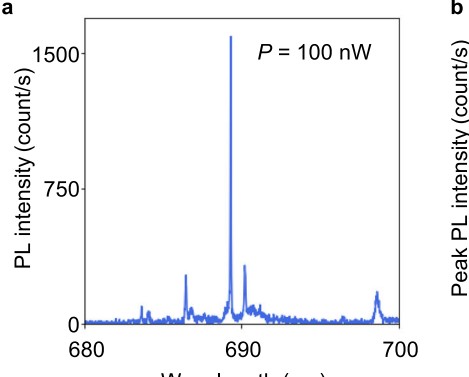

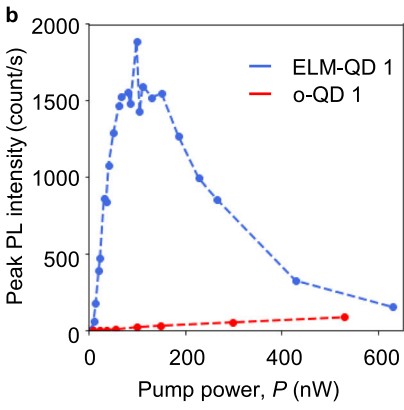

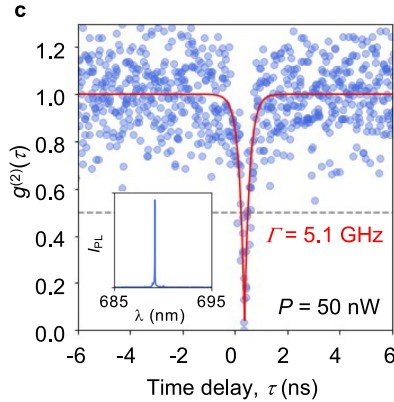

**Fig. 4 | Single-photon emission from ELM-QDs. a** PL spectrum of the ELM-QD in Fig. 3a (ELM-QD 1) pumped with the excitation power of 100 nW. **b** Pump power dependence of the peak PL intensity from ELM-QD (blue) and o-QD (red). **c** Second-order correlation function, $g^{(2)}(\tau)$, for photon emission from ELM-QD 1. Data points are fitted with an exponential decay function, $y(\tau) = 1 - A \exp(-\Gamma\tau)$, where $A = 0.96$ and. $\Gamma = 5.1$ GHz: Inset: filtered ELM-QD PL spectrum.

quantum confinement energy, therefore, provide a tuning knob for engineering the attractive potential near LDE QDs (Supplementary Note 3).

Next, we explore whether the modified energy landscape can be exploited for on-demand electrical pumping. As a step toward this goal, we consider remote optical excitation when a QD is not pumped directly but at a distance, with the photoexcited charge carriers requiring travel of micrometers to reach the QD. Such remote optical pumping could also reduce dephasing due to thermal fluctuations caused by absorption of the pump beam. To investigate the potential for remote optical excitation, we monitored PL from ELM-QDs while exciting them with a focused optical beam at a variable distance, up to several micrometers away. Fig. 3d summarizes the intensity of QD exciton emission as a function of the distance between the QD and the pump beam position. The ELM-QD continues emitting photons even when the pump beam is displaced ~5–10 μm away from the QD location, showing effective carrier funneling due to the modified energy landscape. In contrast, PL intensity for o-QDs drops dramatically when the pump beam is displaced from the dot by only 1–2 μm, a distance comparable to the width of the point-spread function in our PL imaging system.

The charge-carrier funneling effect is most clearly evident in an increase in the overall efficiency for ELM-QDs compared to the o-QD, and it is summarized in Fig. 4, showing the PL intensity for the QDs excited at different optical pump powers. For the ELM-QD, a narrow linewidth peak emission at 1.80 eV (689 nm) increases linearly with the pump power first, then, at ~100 nW, the peak intensity saturates and starts decreasing, the linewidth broadens and additional lines due to exciton complexes appear[49–52] (see Supplementary Note 5). In contrast, the o-QD shows a gradual increase in PL intensity in the same range of the pump powers. In Fig. 4b, PL from the o-QD 1 is two orders of magnitude weaker than that from ELM-QD 1 at the pump power of 100 nW. PL intensity for ELM-QDs on average is one order of magnitude higher compared to that for o-QDs at low excitation powers (see Supplementary Note 5), and the saturation power for the ELM-QDs is on average over one order of magnitude lower, indicating an increased overall efficiency due to more efficient excitation. We also find that the excitation efficiency $\eta_x$ for the ELM-QDs is in the range of 10–170 times higher than that for previously reported o-QDs[13,39,40,53–58] (detailed calculations are provided in Supplementary Note 6). We note that this level of efficiency improvement is comparable to the outcoupling improvement achieved with photonic environment engineering[59] (Supplementary Note 6).

Most importantly, we found that the funnel preserves the single-photon emission properties of GaAs QDs located in it while enhancing excitation efficiency under non-resonant excitation. Using a Hanbury-Brown-Twiss interferometer, we determined that the ELM-QD in Fig. 3a, optically pumped at 50 nW displayed a SPE signature with a fast decay rate of 5.1 GHz (Fig. 4c). The decay rate $\Gamma$ is extracted by fitting an exponential

decay function, $y(\tau) = 1 - A\exp(-\Gamma\tau)$, to the second-order correlation function data. The faster decay rate compared to previously reported values[43] is likely to originate from readily available charge carriers in the funnel, around the QD. The minimum of the second-order correlation function, $g^{(2)}(0) = 0.04$ was determined by including the random coincidence correlation correction[60] (Supplementary Note 7). The $g^{(2)}(0)$ measurement satisfies the criterion for SPE ($g^{(2)}(0) < 0.5$), and therefore, the efficiency of GaAs LDE QDs is increased without destroying their SPE properties. We note that the $g^{(2)}$ measurements were performed using a pair of edge filters (see Methods) with total transmission at the QD wavelength of ~25%, limiting the overall efficiency.

In conclusion, we introduce and demonstrate a charge-carrier funnel for enhancing the overall quantum efficiency of GaAs LDE QDs under non-resonant optical excitation. We engineered the in-plane energy landscape around the QD by leveraging the specially-grown microscale AlGaAs disks, which attract and funnel excited charge carriers to the QDs, resulting in a significant increase in PL efficiency—over one order of magnitude greater than that of ordinary LDE GaAs QDs. These funnels can serve as effective charge-carrier attractors and reservoirs, enabling remote, up to 10 μm, optical excitation and promising an efficient and scalable material platform for electrically driven QD SPEs[25]. This energy landscape engineering approach provides a practical strategy to enhance excitation efficiency and reduce the saturation pump power. Further investigations, however, are needed to evaluate its impact on SPE metrics such as brightness, photon indistinguishability and purity. We anticipate that the combination of the efficient charge-carrier delivery via energy landscape engineering with photonic environment engineering and electrical charge injection will open new avenues for developing scalable electrically driven quantum light sources that can be integrated into future quantum information systems[27,42,61,62].

## Methods
### Material growth
GaAs QD samples were grown by MBE on a GaAs (100) substrate. The epitaxial structure consists of (from bottom) a 300 nm thick GaAs smoothing layer, a 500 nm $Al_{0.75}Ga_{0.25}As$ sacrificial layer and a 140 nm $Al_{0.4}Ga_{0.6}As$ barrier layer protected on both sides with 5 nm thick GaAs layers, all grown at 600 °C. Charge-carrier funnels were formed from gallium droplets, which were deposited throughout the growth. The probability of gallium droplet formation is higher for lower temperatures of the tip of the gallium effusion cell[63], and the tip:base temperature ratio of the cell was varied to favor the droplet formation. Droplets deposited ~20–30 nm into the growth of the $Al_{0.4}Ga_{0.6}As$ barrier from the charge-carrier funnels. A sheet of low-density GaAs LDE QDs was embedded in the middle of the barrier layer. First, the growth process was paused and the substrate temperature was increased to 620 °C under an arsenic soak; the excess arsenic on

the surface was removed by annealing the substrate at this temperature for 40 s without any arsenic supply. Next, only aluminum was introduced to form droplets with a nominal thickness of 0.6 ML. Then the droplets were annealed in a low-arsenic environment for 300 s to promote etching of nanovoids. To form QDs in the voids, GaAs was deposited using migration-enhanced epitaxy followed by annealing for 300 s. The Ga and Al growth rates were in the range of 0.4–0.5 ML/s. The As:Ga beam equivalent pressure ratio was maintained at 45–50 for most of the growth and reduced to ~7.5 for a low-arsenic environment.

## PL spectroscopy and imaging

Samples were cooled down to 10 K in the closed-loop Montana Instruments cryostat. A 520 nm wavelength laser diode was used for non-resonant optical excitation. The laser beam was focused using a microscope objective with a numerical aperture of 0.4. The laser beam spot diameter was ~1 μm. PL was collected through the same microscope objective and analyzed using a 50 cm length grating spectrometer equipped with 600 and 1800 grooves per millimeter gratings and a $1340 \times 100$ back illumination array detector (Teledyne Princeton Instruments). A pair of superconducting nanowire single-photon detectors (Quantum Opus) and a time-correlated single-photon counting module (PicoHarp 300, PicoQuant) were used for measuring the second-order correlation function. In these measurements, the emission from the QD was isolated using a pair of long-pass and band-pass filters (Supplementary Note 7), with the total transmission of ~25% at the QD wavelength. PL maps and correlation time tagging measurements were collected using pyscan (github.com/sandialabs/pyscan), an open-source measurement tool box developed at the Center of Integrated Nanotechnologies.

## Band energy calculation

We calculated the energy profile in Fig. 2e using NextNano, a commercial Schrodinger-Poisson equation solver. To determine the conduction band energy profile within the funnel, we used the quantum well (QW) model and calculated the electron confining energy in an $Al_{0.4}Ga_{0.6}As/Al_{0.2}Ga_{0.8}As/Al_{0.4}Ga_{0.6}As$ QW, where the $Al_{0.2}Ga_{0.8}As$ layer thickness was assumed to vary as a function of position $x$, according to the epilayer height profile found in AFM measurements (Fig. 2d, e), while the material composition was determined from EDS profiles in Fig. 2c.

## Data availability

All data presented in this Article have been deposited in the Figshare Repository (https://doi.org/10.5522/04/30437846).

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

## Acknowledgements

Research was supported by the U.S. Department of Energy (DOE), Office of Science, Basic Energy Sciences (BES), under Field Work Proposal Number 24–017574, and performed, in part, at the Center for Integrated Nanotechnologies, an Office of Science User Facility operated for the U.S. DOE Office of Science. S.Pr. acknowledges support by the EPSRC (EP/S022139/1). A.M. acknowledges support by the Laboratory Directed Research and Development (LDRD) Nanodevices and Microsystems Program. Sandia National Laboratories is a multi-mission laboratory managed and operated by National Technology and Engineering Solutions of Sandia, LLC., a wholly owned subsidiary of Honeywell International, Inc., for the U.S. DOE's National Nuclear Security Administration under contract DE-NA0003525. This paper describes objective technical results and analysis. Any subjective views or opinions that might be expressed in the paper do not necessarily represent the views of the U.S. DOE or the United States Government.

## Author contributions

Conceptualization—S.Pa., S.A., P.P.I, I.B., O.M.; methodology—S.A., P.P.I, I.B., O.M.; Experimental investigation, sample preparation, simulations and data analysis—S.Pa., D.S., G.B., J.Y., H.J., S.Pr., P.C., S.A., D.P., P.P.I., I.B., O.M.; experimental system development—S.Pa., K.M.A.-R., J.J.M., A.M., T.S.L., P.P.I., O.M.; project administration and supervision, I.B., O.M.; writing, original draft—S.Pa., S.A., P.P.I, I.B., O.M; writing, review and editing—all authors.

## Competing interests

The authors declare no competing interests.
