## [Transparent Peer Review file · Communications Materials]

Efficient single-photon emission via quantum-confined charge funneling to quantum dots

Corresponding Author: Dr Oleg Mitrofanov

This manuscript has been previously reviewed at another journal. This document only contains information relating to versions considered at Communications Materials.

Version 0:

Decision Letter:

** Please ensure you delete the link to your author homepage in this email if you wish to forward it to your coauthors **

Dear Dr Mitrofanov,

Thank you once again for submitting your manuscript, "Efficient single-photon emission via quantum-confined charge funneling to quantum dots," to Communications Materials. It has now been seen again by the referees, whose comments are appended below. The concerns of our reviewers have now been largely addressed, but there are some amendments needed before we can accept your paper.

In particular, you will see that Reviewer #2 has some final requests, and we ask that you accommodate these in the final version of the paper. Please include a point-by-point reply to these remarks, for the editor to check.

We ask that you edit your manuscript according to the attached table. **Please read this document carefully as we will be unable to further assess your revised paper until these important points are addressed.**

Please outline all revisions made in the right-hand column and return the completed table with your updated manuscript files as a Related Manuscript file.

Please use the link below to submit your revised files:

Link Redacted

When resubmitting, please provide a marked-up manuscript with all changes highlighted, as well as a clean version of your paper.

We hope to receive this updated version of your paper within 1 week, but please let us know if you find that you need more time.

Best regards,

Dr Aldo Isidori
Senior Editor
Communications Materials

Reviewers' comments:

Reviewer #1 (Remarks to the Author):

I appreciate all the amendments and corrections, as well as the extension of the supplementary information. I also value the effort the authors invested in the rebuttal letter to justify their points and answer questions from all reviewers. I am satisfied with the current status of the manuscript, which presents a proof-of-principle study that is scientifically interesting and opens

avenues for follow-up works addressing the aspects mentioned in the first review round that are (still) missing here. I support its publication in Communications Materials.

Reviewer #2 (Remarks to the Author):

This revision addresses the main concerns of the reviewers: claims are toned down, figures clarified, and efficiency analysis added. The Abstract sentence "This has hindered their applications in quantum information systems, including for multi-photon cluster state generation and Boson sampling" should still be softened, as it over-links non-resonant inefficiency to those applications. I suggest: "This low excitation efficiency under non-resonant pumping limits overall source efficiency; here we address this bottleneck with a charge-carrier funnel." With this adjustment, the work is suitable for publication.

Version 1:

Decision Letter:

Dear Dr Mitrofanov,

We are delighted to accept your manuscript titled "Efficient single-photon emission via quantum-confined charge funneling to quantum dots" for publication in Communications Materials. Thank you for choosing to publish your interesting work with us.

Licence to Publish and Article-Processing Charge

In approximately 7-10 business days you will receive an email with a link to choose the grant of rights necessary for publishing your paper and – if applicable – to provide payment information for your article-processing charge (APC), either via credit card or by requesting an invoice.

If needed, our Author Services team will be in touch regarding any additional information that may be required.

In order to avoid any delays, please ensure that you have emails from Springer Nature whitelisted in your mail system.

We will edit your manuscript to ensure that it conforms with our house style and send you a link to an online eProof for checking in a separate email to the publishing agreements. Please read your proof with great care to ensure that the sense has not been altered. We also suggest you discuss the proof with your co-authors, but please ensure that only one author communicates with us and that only one set of corrections is returned via the online correction in the eProof. The corresponding (or nominated) author is responsible on behalf of all co-authors for the accuracy of all content, including spelling of names and current affiliations.

To ensure prompt publication, your proofs should be returned within two working days. If there is any period within the next four weeks in which you won't be available, please nominate a co-author with whom we can correspond, and let us know their e-mail address as soon as possible.

Please note that production will not continue until the Licence to Publish and Article-Processing Charge steps are completed and your proof corrections are submitted.

Please note that your Supplementary Information files are now finalized. They will be uploaded directly to the Communications Materials website in preparation for publication of the Article. Any requests to make changes will only be considered in exceptional circumstances and will result in a delay to publication.

Acceptance of your manuscript is conditional on all authors' agreement with [our publication policies](https://www.nature.com/commsmat/editorial-policies). In particular, your manuscript must not be published elsewhere and there must be no announcement of the work in the media until the publication date. At this stage, you may wish to make your institution's press office aware of the forthcoming publication, if you wish to bring your work to the media's attention, so that they can start preparing any publicity. Please note that the paper is still under embargo until it is published in the journal. Further details of our embargo policy can be found here <http://www.nature.com/authors/policies/embargo.html>.

We will aim to publish your article in a timely manner. Please note there will be no further correspondence about your publication date. When your article is published, you will receive a notification email. **If you are planning an embargoed press release or require a specific publication date, please complete our form**

<https://forms.office.com/e/ed7NBDd08u>>scheduling requests form, or contact **commsproduction@springernature.com**, as soon as possible after acceptance and we will endeavour to accommodate your request. For further information on the journey of your article from acceptance to publication, please see our [Author FAQs](https://www.nature.com/documents/Author_FAQs.pdf).

If you have any questions about open-access invoicing or payment, please contact authororders@nature.com

Best regards,

Dr Aldo Isidori
Senior Editor
Communications Materials

***As a new journal, we would greatly appreciate any comments you have about your experience at Communications Materials. I hope that we have been able to meet your expectations and look forward to working with you again in the future.

We may promote your article on social media once it is published, so please feel free to send me the twitter handles of any authors or departments and we will be sure to tag them accordingly.***

Response to Reviewers

We appreciate the time the Reviewers dedicated to reviewing our Revised manuscript. We are pleased that the Reviewers found their original concerns have been largely addressed. We also appreciate the final additional suggestion from Reviewer 2 to modify the abstract. Below, we describe point-by-point the amendments requested by Reviewer 2, as well as amendments requested by the editorial office.

For clarity we used color-codes in this Response to Reviewers letter:

Our response is shown in black

Reviewers' comments are shown in dark red

Modified text is shown in *Italics blue*

Reviewer #1 (Remarks to the Author):

I appreciate all the amendments and corrections, as well as the extension of the supplementary information. I also value the effort the authors invested in the rebuttal letter to justify their points and answer questions from all reviewers. I am satisfied with the current status of the manuscript, which presents a proof-of-principle study that is scientifically interesting and opens avenues for follow-up works addressing the aspects mentioned in the first review round that are (still) missing here. I support its publication in Communications Materials.

Reply 1: We thank the Reviewer for his/her thorough review of the Manuscript, Response to Reviewers Comments and Supplementary information. We value his/her comments and suggestions throughout the review process.

Reviewer #2 (Remarks to the Author):

This revision addresses the main concerns of the reviewers: claims are toned down, figures clarified, and efficiency analysis added. The Abstract sentence "This has hindered their applications in quantum information systems, including for multi-photon cluster state generation and Boson sampling" should still be softened, as it over-links non-resonant inefficiency to those applications. I suggest: "This low excitation efficiency under non-resonant pumping limits overall source efficiency; here we address this bottleneck with a charge-carrier funnel." With this adjustment, the work is suitable for publication.

Reply 2: We thank the Reviewer for his/her time reviewing our original and revised manuscript, and our Response letter. We also appreciate the final suggestion to amend the Abstract.

The Reviewer suggested to soften one sentence in the Abstract, and we have removed that sentence following their recommendation. Furthermore, the Reviewer suggested to emphasize that the manuscript addresses the inefficiency of non-resonant pumping, and we added 'under non-resonant pumping' in the following sentence in the Abstract.

The amended sentences from the Abstract are shown below in highlights:

However, their overall quantum efficiency – determined by a product of the internal, excitation, and out-coupling efficiencies – remains limited, primarily due to low (typically below 0.1%) excitation efficiency. ~~This has hindered their applications in quantum information systems, including for multi-photon cluster state generation and Boson sampling.~~ To mitigate the low excitation efficiency under non-resonant pumping, here we realize liquid droplet etched GaAs QDs in a microscale 3D AlGaAs charge-carrier funnel.